# Locoregional Therapy for Intrahepatic Cholangiocarcinoma

**DOI:** 10.3390/cancers15082384

**Published:** 2023-04-20

**Authors:** Mackenzie Owen, Mina S. Makary, Eliza W. Beal

**Affiliations:** 1The Ohio State University College of Medicine, Columbus, OH 43210, USA; 2Division of Vascular and Interventional Radiology, The Ohio State University Wexner Medical Center, Columbus, OH 43210, USA; 3Departments of Surgery and Oncology, Barbara Ann Karmanos Cancer Institute, Wayne State University School of Medicine, Detroit, MI 48201, USA

**Keywords:** intrahepatic cholangiocarcinoma, locoregional therapy, radiofrequency ablation, microwave ablation, transarterial chemoembolization, transarterial radioembolization, external beam radiotherapy, stereotactic body radiotherapy, hepatic arterial infusion, irreversible electroporation

## Abstract

**Simple Summary:**

Intrahepatic cholangiocarcinoma is an aggressive primary liver cancer originating in the intrahepatic bile ducts. While surgical resection is the only curative treatment, many patients present with locally advanced, unresectable, or metastatic disease, and few are candidates for curative-intent resection. In this review, we examine locoregional therapy approaches and summarize the current literature. Current locoregional therapies include thermal ablation, transarterial chemoembolization, transarterial radioembolization, external beam radiotherapy, stereotactic body radiotherapy, hepatic arterial infusion of chemotherapy, irreversible electroporation, and brachytherapy. These therapies are most often offered to patients with unresectable primary or recurrent intrahepatic cholangiocarcinoma, and studies on each modality have shown these locoregional approaches to be effective for prolonging overall survival. The findings of this review also further inform the need for future research regarding the efficacy of these treatments in comparison to each other due to the limited literature on optimal treatment strategies.

**Abstract:**

Intrahepatic cholangiocarcinoma (ICC) has a poor prognosis, and surgical resection (SR) offers the only potential for cure. Unfortunately, only a small proportion of patients are eligible for resection due to locally advanced or metastatic disease. Locoregional therapies (LRT) are often used in unresectable liver-only or liver-dominant ICC. This review explores the role of these therapies in the treatment of ICC, including radiofrequency ablation (RFA), microwave ablation (MWA), transarterial chemoembolization (TACE), transarterial radioembolization (TARE), external beam radiotherapy (EBRT), stereotactic body radiotherapy (SBRT), hepatic arterial infusion (HAI) of chemotherapy, irreversible electroporation (IE), and brachytherapy. A search of the current literature was performed to examine types of LRT currently used in the treatment of ICC. We examined patient selection, technique, and outcomes of each type. Overall, LRTs are well-tolerated in the treatment of ICC and are effective in improving overall survival (OS) in this patient population. Further studies are needed to reduce bias from heterogenous patient populations and small sample sizes, as well as to determine whether certain LRTs are superior to others and to examine optimal treatment selection.

## 1. Introduction

Intrahepatic cholangiocarcinoma (ICC) is a rare and aggressive primary hepatic malignancy with increasing incidence in the United States (US) and worldwide. Between 1973 and 2012, the incidence of ICC has risen from 0.4 to 1.18 cases per 100,000 persons in the USA and represents the second most common primary liver cancer behind hepatocellular carcinoma (HCC) [1]. ICC carries a poor prognosis with a 5-year OS of less than 10% and increasing mortality rates [1,2]. The only curative treatment for ICC is surgical resection (SR), though only up to 30% of patients are eligible because the disease is often locally advanced or metastatic at the time of presentation [3]. Even with SR, the 5-year OS remains low at 22–45%, and recurrence rates are as high as 80% [4]. For patients who are able to undergo SR, adjuvant capecitabine is recommended based on the results of the BILCAP trial, which demonstrated improved OS for adjuvant capecitabine in the per-protocol analysis [5]. Locoregional therapies (LRTs) have also been compared to SR for the primary treatment of early-stage disease or used in the adjuvant setting, after SR.

For patients with unresectable or metastatic disease, clinical trials, systemic therapy, chemoradiation (ChR), and LRTs are among the treatment options [6]. On the basis of the ABC-02 trial, gemcitabine and cisplatin doublet therapy are the preferred regimens for systemic therapy in the first-line setting [7]. Triplet therapy with durvalumab, gemcitabine, and cisplatin was also recently adopted as a preferred option in the first-line setting after demonstrating improved OS compared to placebo plus chemotherapy in the TOPAZ-1 study [8]. FOLFOX is the preferred regimen in the second-line setting for patients who progress on gemcitabine and cisplatin [9].

For patients with unresectable ICC with a liver-only or liver-predominant disease, LRTs represent a promising treatment option for multimodality treatment [6,10]. LRTs currently in use for ICC include RFA, MWA, TACE, TARE, EBRT, SBRT, HAI, IE, and brachytherapy. Preliminarily, these LRTs have shown promising results in the treatment of ICC, with improved survival rates [11]. The objective of this review is to discuss the LRTs currently in use and to summarize the current literature.

## 2. Methods

In this narrative review, we summarized articles related to locoregional therapies for intrahepatic cholangiocarcinoma. We included articles published between 2005 and 2022. We included articles with data on outcomes from the following locoregional therapies: RFA, MWA, TACE, TARE, EBRT, SBRT, HAI, IE, and brachytherapy. Articles were excluded if they focused on extrahepatic cholangiocarcinoma including distal or hilar cholangiocarcinoma, or gallbladder cancer, or if they included patients with all biliary tract cancers and the data for patients with intrahepatic cholangiocarcinoma could not be separated.

## 3. Locoregional Treatment for Intrahepatic Cholangiocarcinoma

### 3.1. Radiofrequency Ablation

#### 3.1.1. Patient Selection

RFA is an LRT that can treat a variety of solid tumors, with predominant use in the liver. For ICC specifically, RFA represents a promising option for patients who are not candidates for curative SR due to advanced cancer at diagnosis, poor hepatic reserve, or serious comorbidities [12,13]. RFA also represents a treatment option for patients who have recurrence after SR [14,15]. In examined studies, exclusion criteria for RFA in ICC varied, though generally patients with severe coagulopathy, severe thrombocytopenia, vascular invasion, tumor size > 5–7 cm, multiple hepatic lesions > 3–5, progressive extrahepatic metastases, or poor performance status were excluded [13,16,17,18].

#### 3.1.2. Technique

RFA is a minimally-invasive technique that is most commonly performed percutaneously by an interventional radiologist with the patient under general anesthesia utilizing imaging-guidance including ultrasound (US) and computed tomography (CT) [18,19]. Procedural characteristics, as well as outcomes, are summarized in Table 1. RFA can also be performed in an open fashion, intraoperatively, and/or in conjunction with SR [20]. The technique utilizes a number of needle electrodes, depending on tumor size and location, often set at a 200 W current for a range of 10–90 min to ablate visible tumors with margins of 5–10 mm [16,21]. The high frequency of the electric current emitted by the electrode generates frictional heat, which causes localized cell death [22]. Technical success, defined as the treatment of the tumor according to protocol, has been reported at rates ranging from 80–100% [23]. Patients are often followed with multiphasic CT or magnetic resonance imaging (MRI) to assess for imaging response, typically obtained 1 month after the procedure [23]. Technical effectiveness based on complete imaging response at 1 month similarly ranged from 80–100% [21]. Lower rates of effectiveness were observed in patients with larger tumors, often >5 cm [12,24]. Local recurrences were often treated with repeat RFA [16,19,25,26]. 

#### 3.1.3. Outcomes

Studies have shown that RFA is safe and well-tolerated in the treatment of ICC. Complications were variably defined by the study, but described major complications included: liver abscess, symptomatic pleural effusion, biloma, biliary stricture, and intrahepatic hematoma. Major complications were rare, occurring at a rate of 0–13% [12,13,14,15,16,18,20,24,25,26,27,28,29,30]. On the other hand, minor complications consisted primarily of post-ablation syndrome as well as asymptomatic pleural effusions or hematomas, bile duct dilation, or gallbladder wall thickening [12,13,14,15,16,18,20,24,28,29,30]. Minor complications resolved without treatment or prolongation of hospital stay. The duration of hospitalization was not commonly reported but three studies reported short hospital stays of 2–4 days in patients without complications [13,16,19]. Similarly, the impact of RFA on liver function was seldom reported, though two studies mentioned transient elevations in transaminases that resolved within months of the procedure [13,19].

The studies examined in this review contained varying patient populations, including those with early-stage ICC to those with recurrent, unresectable ICC. For patients with unresectable or recurrent ICC treated with RFA, median OS ranged from 20–60 months, compared to median OS rates of 3–8 months in patients with unresectable ICC who did not undergo any treatment [6,12,13,14,15,17,18,20,26,27,28,29,32]. Interestingly, Xiang et al. found that SR showed significantly improved 1-, 3- and 5-year OS and cancer-specific survival (CSS) rates compared to RFA for stage I tumors < 5 cm [17]. On the other hand, Wu et al. found that RFA conferred a significant 5-year OS benefit for stage I tumors < 5 cm compared to ChR with an OS rate of 20.1% compared to 3.7%, respectively [32]. Overall, these results are encouraging, though further randomized controlled trials with larger sample sizes and prospective designs are warranted.

### 3.2. Microwave Ablation

#### 3.2.1. Patient Selection

Patient selection for use of MWA in ICC patients is similar to that of RFA. Studies examined in this review used MWA for primary unresectable ICC, recurrent ICC after SR, and in the case of larger tumors, with palliative intent (Table 2). Many studies included only patients with a largest tumor size of <5 cm and <3 lesions, though some studies included patients with larger tumors, up to 10 cm [25,26,33,34,35,36,37,38,39,40,41,42,43,44]. Most eligible patients were required to have sufficient coagulation parameters, platelet count, liver and kidney function, and adequate performance status [25,33,34,36,39,40,42,43,44]. Patients with major vascular invasion by the tumor, extrahepatic metastases, acute severe infection, recurrent ascites, and other biliary tract or hepatic tumors were generally excluded from treatment with MWA [25,26,33,36,37,38,39,40,41,42,43,44].

#### 3.2.2. Technique

MWA is a minimally-invasive technique most commonly performed with patients under general anesthesia while a microwave probe is inserted percutaneously into the tumor under imaging-guidance [33]. MWA emits electromagnetic radiation to induce tumor cell death via frictional heat, though is capable of generating higher temperatures in shorter time periods, with the goal of more completely ablating targeted tissue and avoiding nearby structures [22]. In contrast to RFA, MWA utilizes uniform heating, has more predictable ablation zones that can target larger liver volumes, and can treat multiple lesions simultaneously, making it a frequently used modality [45]. Included studies used a variety of settings for MWA. Most commonly, the output power was set to 40–100 W for 3–20 min [25,33,34,36,37,39,41,42]. Single electrodes were often used for tumors < 2–3 cm, while multiple electrodes or ablations were used for larger tumors [25,33,34,37,41,42,43,44]. Tumors were treated with the goal of achieving 0.5–1 cm margins, and the needle tract was ablated during the removal of the probe to avoid tumor seeding [25,26,34,38,39,42,43]. Technical success was defined as the ability to treat the tumor according to protocol and was assessed 2–5 days after treatment with contrast-enhanced (CE) CT or MRI [26,33,34,39,43,44]. Technical effectiveness was defined as complete ablation 1 month after the procedure, and most studies describe subsequent ablations if a residual tumor was present [36,38,40,43,44]. After an initial 1-month follow-up, patients generally followed up every 3–6 months [25,33,34,36,37,38,39,40,41,42,43].

#### 3.2.3. Outcomes

Like RFA, studies have shown that MWA is well-tolerated in the treatment of ICC. Major complications ranged from 0–20% in examined studies and consisted of liver abscess, pleural effusion, ascites, hepatic failure, and tumor seeding [25,26,35,36,38,39,40,41,42,43,44]. Minor complications were common and included post-ablation syndrome, thrombocytopenia, a transient elevation in liver function tests (LFTs), and portal vein thrombosis [25,39,42,44]. Giorgio et al. reported that hospital stays ranged from 1–4 days [38].

Survival data varied widely between studies, likely due to the variable patient populations included in each study. Of studies examined in this review, the median OS of ICC patients treated with MWA ranged from 8.8–31.5 months, and median progression-free survival (PFS) ranged from 6.2–18.43 months [25,26,33,35,36,40,41,42,43,44]. Interestingly, the study by Yan et al. assessed the impact of combining thermal ablative therapy, consisting of RFA or MWA, with systemic chemotherapy, compared to chemotherapy alone in the treatment of unresectable and previously untreated ICC. They found that the median OS was significantly higher in the combined group (combined median OS = 15.23 months vs. chemotherapy alone = 7.97 months, *p* = 0.009) [34]. Additionally, the study by Giorgio et al. was unique in that it directly assessed OS and PFS in patients with unresectable ICC treated with RFA vs. MWA [38]. This study demonstrated a statistically significant increase in both OS and PFS favoring MWA [38]. Finally, the study by Xu et al. was notable for assessing median OS and complication rates in patients with recurrent ICC after initial SR, treated with MWA vs. repeat SR [40]. They found no significant difference in median OS between the two groups, but the repeat SR group had a statistically significant increase in major complications (MWA = 5.3% vs. repeat SR = 13.8%, *p* < 0.001) [40]. While data between studies are heterogeneous, results show that MWA is both well-tolerated and similarly effective in comparison to other LRTs.

### 3.3. Transarterial Chemoembolization

#### 3.3.1. Patient Selection

TACE is an intra-arterial therapy that has been employed in various hepatic malignancies, including ICC. While TACE is a non-curative therapy, it represents an option for locoregional tumor control in patients ineligible for SR, as an adjuvant therapy after SR, or in patients with progression of disease after initial treatment [46,47]. Inclusion and exclusion criteria varied between studies, and eligible patients generally required adequate performance status as well as sufficient bone marrow, liver, and renal function [48,49]. Patients with severe comorbidities, active infection, or contraindication to arterial procedures were generally excluded [48,49]. Tumor size was not mentioned as an exclusion criterion in the examined studies for TACE. This is in contrast to RFA, where patients with tumors >5 cm were often considered ineligible.

#### 3.3.2. Technique

TACE represents a treatment option for patients with locally advanced tumors, with the goals of delivering higher concentrations of chemotherapeutic agents locally and engendering tumoral ischemia [50]. The technique, outcomes, and complications for TACE in included studies are summarized in Table 3A. During the procedure, the hepatic artery supplying the tumor is most commonly accessed via a femoral approach and identified with conventional angiography [48,49,51]. In conventional TACE (cTACE), a chemotherapeutic agent emulsified with lipiodol is injected into the hepatic artery followed by arterial occlusion with embolic material, depriving the tumor of blood supply [51]. Commonly used chemotherapies include doxorubicin, cisplatin or carboplatin, mitomycin-C, and gemcitabine [46]. In a slightly different procedure, drug-eluting bead TACE (DEB-TACE) employs chemotherapy-laden beads or microspheres to both deliver the medication and embolize the artery simultaneously [49,52]. In both cTACE and DEB-TACE, the procedure is complete when near-stasis is achieved on angiography. Most studies assessed tumor response to treatment with follow-up imaging using CT or MRI according to the modified Response Evaluation Criteria in Solid Tumors (mRECIST) criteria [48,49,52,53,54]. Repeat sessions of TACE can be performed for residual tumors, recurrence, or progression of the disease.

#### 3.3.3. Outcomes

Similar to RFA, studies have shown that TACE is a safe and well-tolerated treatment method for ICC. Major complications were rare but included inguinal hematoma, hepatic arterial dissection, hepatorenal syndrome, severe thrombocytopenia, and hepatic abscess [53,55,56]. Major complications occurred at a rate of 0–12.5% [3,49,52,53,55,57,61]. Minor complications included post-embolization syndrome, which consists of abdominal pain, fatigue, nausea, vomiting, and fever, as well as transient decreases in liver function [61]. Luo et al. and Poggi et al. noted that elevations in liver transaminases were brief and rapidly returned to baseline in 1–3 months after TACE whereas Sun et al. described no difference in liver function before and after treatment [49,53,57]. Duration of hospital stay after TACE was not commonly reported in the included studies; however, Kim et al. noted a median hospital stay of 4 days [56].

Data assessing OS after TACE is limited and included studies vary significantly in patient selection, sample size, and study design. However, included studies reported OS rates ranging from 6–30 months [3,48,49,50,52,53,57,61]. Interestingly, Sun et al. reported a significant increase in OS for 40 patients treated with DEB-TACE compared to 49 patients treated with cTACE at a median of 10 months versus 6 months, respectively (*p* = 0.006) [53]. Finally, Hu et al. compared chemotherapy plus DEB-TACE to chemotherapy plus cTACE to chemotherapy alone with median OS of 19.3, 14.0, and 6.5 months, respectively [48]. While these results are promising, again, larger studies and prospective designs are warranted to further assess the efficacy of TACE as an LRT option for ICC.

#### 3.3.4. Adjuvant TACE

Several studies examined TACE specifically in the adjuvant setting after SR (Table 3B) [31,47,58,59,60]. Adjuvant TACE was performed in patients based on the surgeon or patient preference [31,59,60]. Additional inclusion criteria were similar to patients receiving TACE alone, including adequate performance status, bone marrow, and kidney function [31,59,60]. In all included studies, adjuvant TACE was performed 2 months after hepatectomy with subsequent follow-up every 2–3 months [31,58,59,60]. Outcome data were mixed. Studies by Cheng et al. and Li et al. found no significant difference in 1-, 3- and 5-year OS or recurrence rates in patients receiving SR plus adjuvant TACE versus SR alone [58,59]. On the other hand, Wu et al. found a significant difference in OS (*p* < 0.001), favoring adjuvant TACE in patients with poor prognostic factors, defined as TNM stage III or IV, or tumor size > 5 cm [31]. Similarly, Shen et al. found a significant improvement in 1-, 3- and 5-year OS (*p* = 0.045) favoring patients receiving adjuvant TACE, as well as a significant increase in median OS for patients with early recurrence, again favoring the adjuvant TACE group compared to SR alone (12 versus 5 months) [60]. Finally, a systematic review and meta-analysis by Wang et al. found a significant improvement in OS for early-stage ICC patients receiving adjuvant TACE [47]. 

### 3.4. Transarterial Radioembolization

#### 3.4.1. Patient Selection

TARE represents a treatment option for patients with unresectable ICC. Examined studies included patients with primary unresectable ICC as well as recurrent ICC after SR or failure of other treatment modalities, such as chemotherapy-refractory ICC [62,63,64,65,66,67,68,69,70,71,72,73,74,75,76,77,78]. These findings are summarized in Table 4. Eligible patients were required to have sufficient bone marrow, liver, and renal function, as well as good performance status and the ability to tolerate angiography [62,64,65,66,67,68,69,70,71,75,76,77,78]. Ineligible patients were those with significant extrahepatic metastases, significant comorbidities, or other malignancies [62,64,66,67,68,69,77]. The size or burden of the tumor was not commonly an exclusion criterion in the examined studies, and many included patients with tumors > 5 cm or a hepatic tumor burden > 50% [62,63,64,65,68,72,73,76].

#### 3.4.2. Technique

The technique utilized in TARE is similar to that of TACE. Prior to treatment, the hepatic vasculature and pulmonary shunt fraction is assessed via a planning session including diagnostic angiography and administration of 99mTc macroaggregated albumin to the targeted tumoral arterial distribution [62,64,65,66,67,68,69,71,72,75,76,78]. Following pre-treatment mapping, target vessels are injected with yttrium-90 (Y90) resin or glass microspheres in the treatment session, delivering localized radiation doses to the tumor, while sparing nearby normal tissue [62,76,77]. The median administered Y90 activity ranged from 1.5–1.74 GBq in included studies [62,64,65,67,68,73,74,77,78]. Patients were treated with multiple sessions as needed [62,63,65]. Duration of hospital stay after the procedure varied considerably by study, with some protocols discharging patients 2–4 h after the procedure and others discharging patients after 1–4 days [62,70,72,76,78]. Most patients were assessed at 1 month for a response to treatment using the RECIST criteria and CT, MRI, or positron emission tomography (PET) imaging, followed by visits every 3 months [54,62,63,64,65,67,68,70,71,74,75,76,77,78].

#### 3.4.3. Outcomes

Like other LRTs, TARE is well tolerated in the treatment of ICC. With the exception of the study by Edeline et al. which reported that 71% of patients experienced Grade 3–4 adverse effects, all other studies reported Grade 3–5 adverse effects occurring at a rate of 0–24% [62,63,64,65,66,67,70,71,72,73,74,75,76,77,78]. The most commonly reported adverse effects included fatigue, nausea, vomiting, abdominal pain, fever, anorexia, diarrhea, anemia, thrombocytopenia, lymphopenia, and transient elevations in LFTs following TARE [62,63,66,67,70,71,73,74,76,78]. Rarer but more severe complications included gastrointestinal (GI) ulcers, gastritis, pancreatitis, perforated cholecystitis, acute hepatic failure, acute cholecystitis, acute cholangitis, liver abscess and radiation-induced liver disease (RILD) [62,63,66,67,70,71,73,74,75,76,78].

Outcomes for TARE varied widely by study, likely due to small sample sizes and variable patient populations. Median OS ranged from 5.7–33.6 months and median PFS ranged from 2.8–10.1 months [62,63,64,65,66,67,68,69,70,71,72,74,75,76,77,78]. The study by Bargellini et al. was notable for assessing the impact of TARE in three groups: Group A—chemotherapy naïve patients treated with TARE, Group B—patients treated with chemotherapy then adjuvant TARE, and Group C—chemotherapy-refractory patients treated with TARE [64]. The study found no significant difference in OS between the groups and the median OS was 14.5 months [64]. Interestingly, the study by Buettner et al. assessed the impact of using resin versus glass microspheres for TARE, though found no significant difference in median OS (29 months) and median PFS (5 months) between the groups [65]. Finally, the case series by Filippi et al. was unique in assessing the impact of repeat TARE in patients with recurrent ICC after the first TARE [67]. The mean time between the first TARE and recurrence was 7.3 months and the median OS after the first TARE was 16.5 months, though the median OS after the second TARE was unfortunately not reported [67].

### 3.5. External Beam Radiotherapy

#### 3.5.1. Patient Selection

EBRT is an LRT used in the treatment of many cancers and is indicated in ICC as well. Like other LRTs, EBRT is not a curative treatment, though can be used in cases of unresectable ICC, recurrent ICC, as an adjuvant therapy after SR, in combination with chemotherapy, or as a palliative treatment [79,80]. Examined studies are summarized in Table 5. In these studies, few patients were considered ineligible for EBRT. Exclusion criteria consisted of patients with Child–Pugh class C cirrhosis, other primary liver tumors, or other serious conditions [81,82,83]. Tumor size was not a factor in determining eligibility and patients with tumor sizes ranging from 2.2–17 cm were treated in the included studies [84].

#### 3.5.2. Technique

Several different techniques are available for the delivery of EBRT. Studies examined in this review delivered radiation using a linear accelerator with 6-Megavolt (MV) or 15 MV photons or via passive scatter photon beam techniques [82,83,84,88]. The recent study by Smart et al. was unique in that it used hypofractionated photon or proton beams [81]. Before treatment, the size and location of the treatment field were determined by 2-dimensional (2D) or 3-dimensional (3D) CT or MRI imaging [81,82,83,84,88]. Radiation planning also required the determination of gross tumor volume (GTV), clinical target volume (CTV), and planning target volume (PTV). These measures were variably defined by each study and included margins to account for setup error and respiratory movement [81,82,83,84,88]. Median overall radiation dose ranged from 50–58.05 Gray (Gy), most often delivered in daily 2 Gy fractions 5 times a week [81,82,83,84,88]. Notably, Smart et al. treated patients with 15 Gy daily fractions to a median overall dose of 58.05 Gy [81]. Patients were monitored clinically every week during treatment. Response to EBRT was assessed initially at 6 weeks via CT or MRI, followed by monitoring every 3 months [81,82,83,88].

#### 3.5.3. Outcomes

Studies have demonstrated that EBRT is well tolerated. In examined studies, Grade 3 or higher complications occurred at a rate of 12.5% or less [81,82,83,84,88]. Commonly reported adverse effects included neutropenia, thrombocytopenia, elevations in LFTs, anorexia, nausea, vomiting, abdominal pain, fatigue, and fever. Chen et al. reported one case of RILD that resulted in mortality [82]. Smart et al. reported one case of RILD that was treated with glucocorticoids as well as cases of ascites, GI bleeding, hepatomegaly, and the need for biliary intervention after treatment [81]. Tao et al. noted that 5 patients were hospitalized for complications related to EBRT or tumor progression [84]. The need for hospitalization was not mentioned in other studies. Overall, EBRT is safe and well tolerated.

Response to treatment and OS varied significantly between studies because included patient populations and study designs were diverse. Median OS ranged widely from 7–39.5 months [81,82,84,85,86,88,89]. Notably, the retrospective study by Kolarich et al. using the National Cancer Database (NCDB) compared the use of EBRT, RFA, radioactive implants (RI), and no local treatment in nonsurgical ICC patients with stage I-IV disease, and found statistically significant improved median OS for patients with stage I disease receiving EBRT or RFA, stage III disease receiving EBRT or RI, and stage IV disease receiving RI compared to no local therapy [85]. Interestingly, the retrospective cohort study by Hammad et al. using the NCDB examined the impact of adjuvant EBRT after SR and found statistically significant improved OS for patients with R0 resection compared to R1/R2 resections (31.2 vs. 19.5 months, *p* < 0.001); however, after multivariate analysis, adjuvant EBRT was not associated with survival in patients with R1/R2, lymph-node-negative resections [86]. On the other hand, Jackson et al., in another retrospective study using the NCDB, demonstrated improved OS for patients with unresectable, localized ICC receiving EBRT and chemotherapy compared to chemotherapy alone (2-year OS 25.8% vs. 20%, *p* = 0.001) [87]. Finally, Shao et al., in a retrospective study using The Surveillance, Epidemiology, and End Results database (SEER), showed improved OS (*p* = 0.00228) and cancer-specific survival (CSS) (*p* = 0.0037) in patients receiving palliative EBRT compared to no EBRT [80]. As a whole, while results were heterogeneous, these studies indicate that EBRT is an effective treatment for ICC in specific patient populations and especially compared to patients not receiving any LRT.

### 3.6. Stereotactic Body Radiotherapy

#### 3.6.1. Patient Selection

Patient selection for use of SBRT and EBRT in the treatment of ICC is similar. Studies included in this review examined the use of SBRT for patients who had previously received SR with positive surgical margins, as adjuvant therapy after SR, for primary unresectable tumors, locally recurrent disease, or after other previous LRT or chemotherapy (Table 6) [90,91,92,93]. Patients included in these studies required adequate performance status [94]. Exclusion criteria varied by study though consisted of patients with <3–4 lesions, other serious medical conditions, other malignancies, inadequate liver function or volume, inadequate bone marrow function, or prior abdominal radiation [90,94,95,96]. The retrospective study by Sandler et al. limited selection to patients with tumors < 8 cm, though other studies included tumors > 10 cm [94,95,97,98,99,100].

#### 3.6.2. Technique

SBRT is a technique that delivers high radiation doses to a targeted field via 3D or 4-dimensional (4D) image-guided planning [90]. This contrasts with EBRT, which uses 2D X-ray guided techniques and thus is less precise, delivering radiation to both the target and inadvertently, to surrounding tissues [90]. EBRT, though generally well tolerated, has traditionally been limited by adverse effects resulting from the radiation of normal tissues, including RILD and GI bleeding [90,94,102]. Precise planning with SBRT aims to reduce these adverse effects. Prior to SBRT, patients in included studies underwent complete staging workups with CT, MRI, and PET scans [90,92,95,96,97]. For treatment planning, patients underwent CT or MRI imaging, and movement during the respiratory cycle was accounted for by using 4D CT simulation, implantation of fiducial markers, or abdominal compression [90,91,92,93,94,95,96,97]. GTV, CTV, PTV, and internal target volume (ITV) were variably defined by study [91,92,93,94,95,96,97]. Image guidance was used daily during treatment to reduce error [90]. In examined studies, median radiation doses ranged from 30–55 Gy delivered in 1–5 Gy fractions daily, every other day or between chemotherapy cycles, depending on the study [91,92,94,95,96,97,99,100,101]. Patients were followed regularly during treatment [94]. After SBRT, patients followed up initially at 1 month–3 months, followed by every 3–6 months thereafter [92,94,96,97]. Response to treatment was assessed by the RECIST criteria [54].

#### 3.6.3. Outcomes

Similar to EBRT, SBRT is well-tolerated in the treatment of ICC. In included studies, Grade 3 complications ranged from 4.7–53% and Grade 4–5 complications ranged from 0–11% [91,92,94,95,96,97,98,99,100]. The most commonly reported adverse effects were fatigue, nausea, abdominal pain, anorexia, bone marrow suppression, and elevated liver enzymes [91,92,94,95,96,97,100]. More severe complications were rare and included GI bleeding, bowel obstruction, biliary obstruction or stenosis, acute cholecystitis, acute cholangitis, liver abscess, a decline in Child-Pugh class, and liver failure [91,92,94,95,96,97,100]. Interestingly, the rates of Grade 3–5 adverse effects for SBRT in included studies were higher than those for EBRT. This may indicate that despite precise targeting in SBRT, higher radiation doses can still cause complications. However, the interpretation of these data may be complicated by the fact that complications were less frequently reported in included studies for EBRT. Further studies are needed to directly evaluate complication rates for EBRT versus SBRT in the treatment of ICC. Regardless, both are well-tolerated options for LRT.

Outcome data surrounding the use of SBRT for ICC are difficult to interpret due to varied study designs and patient populations. Of examined studies, the median OS ranged from 13.2–23 months [91,94,95,96,97,98,99,100]. PFS ranged from 6.1–24.7 months [91,92,94,95,96,99,100]. These data are challenging to interpret because many of the examined studies included both patients with ICC and extrahepatic cholangiocarcinoma (ECC) and did not separate outcome data between the two diseases [91,92,95,98,99]. This is problematic because studies have shown that these cancers may represent distinct disease processes with different epidemiology, risk factors, genetic makeup, recurrence trends, and response to treatment [103,104,105]. Notably, studies by Kozak et al., Shen et al., Weiner et al., and Tse et al. separated outcome data for ICC specifically and examined the use of SBRT as adjuvant therapy after SR, definitive treatment and for unresectable disease [94,96,97,100]. Finally, the retrospective cohort study by Sebastian et al. was interesting in that it compared the use of SBRT, ChR, and TARE for unresectable ICC [101]. This study found that OS was significantly greater for patients treated with SBRT compared to ChR (*p* < 0.0001) and identified no significant difference in OS between SBRT and TARE [101]. As with other LRTs, more data are needed to examine the impact of SBRT in the treatment of ICC and in comparison to other treatment modalities.

### 3.7. Hepatic Arterial Infusion

#### 3.7.1. Patient Selection

Of studies examined in this review, ICC patients selected for treatment with HAI were those with unresectable, advanced, multifocal, or chemotherapy-refractory tumors [106,107,108,109,110,111,112,113,114,115,116,117,118,119,120,121]. These studies and their results are summarized in Table 7. Patients eligible for HAI generally required good performance status, sufficient hepatic, renal, hematologic, and bone marrow function, adequate nutritional status, and an EKG without actionable abnormalities [107,108,109,112,114,116,119,120,121]. Patients were generally excluded if they had extrahepatic metastases, had a history of portal hypertension, primary sclerosing cholangitis or Gilbert’s disease, had portal inflow occlusion, were unable to tolerate angiography, had an active infection, were pregnant or had other concurrent malignancies [106,107,108,109,110,112,114,116,119,120,121]. Size was not an excluding factor and several studies included patients with tumors >10 cm [107,108,111,113,115,116,118,120,121].

#### 3.7.2. Technique

The goal of HAI is to deliver high doses of chemotherapy directly into the hepatic circulation, thereby producing a more significant local cytotoxic effect than systemic chemotherapies [122]. In most of the studies examined in this review, patients underwent pre-treatment planning with a baseline CT, MRI, or PET scan and angiography to identify the bloody supply to the tumor [107,108,109,110,113,116,120,121]. Patients then underwent percutaneous placement of an infusion pump, port, indwelling catheter, or repeated catheterization for delivery of the chemotherapeutic agent [106,108,109,110,111,112,113,115,116,117,118,119,120,121]. The choice of agent varied by study but the commonly used chemotherapies included floxuridine, gemcitabine, oxaliplatin, 5-fluorouracil, cisplatin, and combinations of these drugs [106,108,109,110,111,112,113,116,117,118,119,120,121]. The number of chemotherapy cycles varied by study and patient but ranged from 1–8, often determined by patient tolerability [106,110,111,115,118,119,121]. Response to treatment was assessed with the RECIST criteria on CT or MRI imaging every 6 weeks to 3 months depending on the study [107,111,114,116,117,118,120].

#### 3.7.3. Outcomes

HAI is well tolerated, similar to other LRTs for ICC. In some of the examined clinical trials, results showed increasing adverse effects with high chemotherapeutic doses [113,119,121]. Overall, however, Grade 3–4 toxicities ranged from 0–22.7% and consisted primarily of nausea, vomiting, fatigue, fever, abdominal pain, diarrhea, and lab abnormalities such as anemia, thrombocytopenia, leukopenia, hyperbilirubinemia, and elevated transaminases [107,108,109,110,113,114,116,117,118,119,121]. Rarer complications included infection, pump misperfusion/embolization, acute pancreatitis, and supraventricular tachycardia [117,121]. The duration of hospital stay ranged from overnight to 6 days after HAI pump placement [106,114].

As with studies on other LRTs for ICC, response to treatment varied by study, likely due to the small sample size and variable patient population. Median OS ranged from 10.1–31.1 months and median PFS ranged from 5–11.8 months [106,107,108,110,111,112,113,114,115,116,117,118,119,120,121]. The study by Franssen et al. was notable for assessing outcomes among patients with multifocal ICC who received either HAI or SR and found no significant difference in median OS, though SR was associated with a significant increase in Grade 3A adverse effects (HAI = 6.4% vs. SR = 25.3%, *p* = 0.04) [106]. Additionally, the study by Ishii et al. found a statistically significant increase in median OS (19.7 months vs. 10.8 months, *p* = 0.006) for patients with advanced ICC receiving HAI vs. systemic chemotherapy [108]. Again, more data are needed to further assess the impact of HAI, especially in comparison to other LRTs.

### 3.8. Irreversible Electroporation

While several studies have examined the use of IE in the treatment of hilar cholangiocarcinoma, data surrounding the use of IE for ICC are extremely limited [123,124,125,126]. Most notably, a prospective study by Belfiore et al. examined the use of IE for unresectable ICC (N = 8) and perihilar (PHCC) cholangiocarcinoma (N = 7) between 2015–2019. In this study, exclusion criteria consisted of cardiopulmonary failure, inadequate hematologic and bone marrow function, contraindications to general anesthesia, bilirubin >3 mL/dL, extrahepatic metastases, recent myocardial infarction, or active infection. Eligible patients underwent pre-treatment staging with CT, endoscopic retrograde cholangiopancreatography (ERCP), and magnetic resonance cholangiopancreatography (MRCP). The procedure was performed under general anesthesia, and tumors were treated with 2–4 bipolar needles or electrodes that delivered 90 pulses at 1500 V/cm, thus ablating the tumor via non-thermal electrical energy. This technique avoids thermal damage to nearby tissues, as seen in other forms of LRT [127]. Response to treatment was assessed in 1-, 3-, and 6-month intervals after the procedure with multidetector CE-CT and MRCP. Belfiore et al. reported no severe complications. The study did not separate ICC and PHCC outcome data; though, via Kaplan–Meier analysis, the mean OS for the entire group was estimated to be 18 mo [127]. In addition to this study, a randomized clinical trial by Zhang et al. assessed the use of IE and cryoablation in ICC and HCC patients, though the trial was designed to assess the impact of allogeneic gamma delta T-cell transfer on outcomes, not IE efficacy [128]. Finally, Eisele et al. included two ICC patients, among several other liver cancer patients, in the use of US-guided IE therapy. The study only assessed local failure rates (21%), not survival outcomes [129]. More studies are clearly needed to assess the use of IE in ICC.

### 3.9. Brachytherapy

Like IE, the literature on the use of brachytherapy for ICC is sparse. One retrospective study by Schnapauff et al. assessed the use of brachytherapy in 15 patients with unresectable ICC without extrahepatic disease between 2006–2009 [130]. Patients with elevated bilirubin >2.5 mL/dL, >5 liver lesions, impaired coagulability and large-volume ascites were excluded from the study [130]. Patients who were eligible underwent treatment planning with contrast liver MRI. Brachytherapy catheters were percutaneously inserted into tumors under CT guidance, with a target CTV of 20 Gy and a permissible dose of >50 Gy in the central parts of the tumor [130]. Brachytherapy delivers high radiation doses to malignant tissue while sparing normal tissue nearby [131]. Unlike thermal ablative options such as RFA and MWA, this treatment has the potential to be effective in tumors with large diameters >5 cm [130]. Patients were followed regularly with tumor markers and liver MRI at 6–12 weeks, followed by 3-month intervals to assess response to treatment [130]. Repeat treatments with brachytherapy were performed in patients with local tumor progression [130]. Tumor size in the study by Schnapauff et al. ranged from 1–18 cm [130]. The study reported a median control time of 10 months after treatment, a median of 13 months to systemic progression and a median OS of 14 months [130]. No severe complications were reported as a result of brachytherapy, though authors acknowledged that the study population size was small [130]. Similar to IE, more studies are needed to determine the feasibility of brachytherapy for the treatment of ICC.

## 4. Conclusions

This review examined the current literature surrounding the use of LRTs for the management of ICC, including RFA, MWA, TACE, TARE, EBRT, SBRT, HAI, IE, and brachytherapy. The majority of studies included in this review were retrospective studies that assessed LRTs for patients with primary unresectable, advanced, or recurrent ICC [66]. Several TACE studies also assessed its use in the adjuvant setting after SR [31,58,59,60]. Across all treatment modalities, patients were generally considered eligible for LRT if they had adequate performance status, sufficient liver, kidney, hematologic, and bone marrow function, and were without significant extrahepatic metastases, comorbidities, and other cancers. RFA and MWA were two treatment modalities where the size of the tumor, often >3–5 cm, was also an excluding factor [22]. The technique used and mechanism of tumor cell death varied by treatment type, though generally LRTs used thermal energy, electric current, chemotherapeutic agents, or radiation delivered locally under image guidance via insertion directly into the tumor in ablative therapies, hepatic circulation in catheter-directed therapies, or externally as in the case of EBRT and SBRT (Table 1, Table 2, Table 3, Table 4, Table 5, Table 6 and Table 7).

LRTs are generally well-tolerated in the treatment of ICC and major complications were rare (Table 1, Table 2, Table 3, Table 4, Table 5, Table 6 and Table 7). Common minor complications included fatigue, nausea, vomiting, diarrhea, abdominal pain, anorexia, and hematologic lab abnormalities. The response to treatment and patient outcomes were variable between LRTs and even among different studies examining the same LRT modality, likely due to heterogeneous patient populations and small sample sizes. Overall, LRTs drastically improve overall survival compared to the natural history of ICC without treatment [6]. Several studies also showed LRTs to be superior to systemic chemotherapy alone [34,57,87,108,111,116]. In regard to the efficacy of LRTs in comparison to each other, few studies assessed this question, and limited conclusions can be made, again due to variable patient population and small sample size [3,37,38,73,101,109,110]. Current National Comprehensive Cancer Network (NCCN) guidelines reflect the importance of pursuing SR for eligible patients and consider LRTs to be suitable options for patients with unresectable or metastatic ICC when part of a clinical trial or at an experienced center [132]. Overall, LRTs are safe and effective options for the treatment of unresectable or recurrent ICC, though more research is needed to assess the superiority of LRT options in comparison to each other.

## Figures and Tables

**Table 1 cancers-15-02384-t001:** Studies examining radiofrequency ablation (RFA) for intrahepatic cholangiocarcinoma.

RFA					
Author (Year)	Study Type and Time Period	Patient Population	Technique	Outcomes	Complications
** Single Cohorts**					
Chu (2021) [27]	Retrospective Study 1999–2019	N = 40 Mean 56.3 y/o Recurrent ICC after curative SR Tumor size < 5 cm	Percutaneous US-guided RFA	Median OS = 26.6 mo OS 3, 5 yr: 36.2%, 18.3%	Major = 4.7%
Brandi (2020) [16]	Retrospective Study 01/2014–06/2019	N = 29 Mean 63 y/o Unresectable ICC Tumor size < 5 cm	Percutaneous US-guided RFA S = median 3	LTPFS = 9.27 mo OS = 27.5 mo	Majorc = 7% Minorc = 14%
Laimer (2020) [19]	Case Report 2007–2019	N = 1 72 y/o Unresectable ICC Tumor size = 10 cm	SRFA S = 10	OS > 11 yr	Majorc = 1× Minorc = 2×
Lee (2020) [20]	Retrospective Study 2009–2016	N = 20 Mean 60 y/o Primary and recurrent ICC with curative intent Tumor size < 3 cm	SR + US-guided IORFA	OS 6 mo, 1, 3, 5 yr: 95%, 79%, 27%, 14% LTPFS 6 mo, 1, 3, 5 yr: 70%, 33%, 13%, 13% Median OS = 22 mo	Major = 5%
Takahashi (2018) [26]	Retrospective Study 02/2006–11/2015	N = 20 Mean 62.5 y/o Primary and recurrent ICC Tumor size < 5 cm	Percutaneous US or CT-guided RFA and MWA	Median DFS = 8.2 mo Median OS = 23.6 mo	Major = 0% Minor = 10%
Butros (2014) [28]	Case Series 01/1998–06/2011	N = 7 Mean 65 y/o Primary and recurrent ICC Tumor size < 5 cm	Percutaneous CT or CT + US-guided RFA S = 1	LTPFS = 36.3 mo OS 1, 3, 5 yr: 100%, 60%, 20%	Major = 0% Minor = 0%
Fu (2012) [13]	Retrospective Study 01/2000–07/2010	N = 17 Median 54.5 y/o Unresectable primary and recurrent ICC Tumor size < 7 cm	Percutaneous US-guided RFA and IORFA S = mean 1.64	OS 1, 3, 5 yr: 84.6%, 43.3%, 28.9% Median RFT = 17 mo Median OS = 33 mo	Major = 3.6% Minor = 47%
Haidu (2012) [18]	Retrospective Study 12/2004–06/2010	N = 11 Median 61 y/o Unresectable ICC Tumor size 0.5–10 cm	Percutaneous SRFA S = mean 2	OS 1, 3 yr: 91%, 71% Median OS = 60 mo	Major = 13%
Xu (2012) [25]	Retrospective Study 10/1998–8/2010	N = 18 Mean 60 y/o Primary and recurrent ICC with curative intent Tumor size < 7 cm	Percutaneous US-guided RFA and MWA	All OS 6 mo, 1, 3, 5 yr: 66.7%, 36.3%, 30.3%, 30.3% Primary ICC OS 6 mo, 1, 3, 5 yr: 87.5%, 75%, 62.5%, 62.5% Median OS = 29.3 mo	Major = 5.5% Minor = 5.5%
Giorgio (2011) [29]	Case Series 01/2003–10/2010	N = 10 Median 70 y/o Unresectable ICC Tumor size 2.4–7 cm	Percutaneous US-guided RFA S = 1–2	OS 1, 3, 5 yr: 100%, 83.3%, 83.3%	Major = 0% Minor = 30%
Kim (2011) [14]	Retrospective Study 10/1999–03/2009	N = 20 Recurrent ICC after curative SR Tumor size 0.7–4.4 cm	Percutaneous US-guided RFA S = mean 1.45	Mean LTPFS = 39.8 mo OS 6 mo 1, 2, 4 yr: 95%, 70%, 60%, 21% Median OS = 27.4 mo	Major = 7% Minor = 55%
Kim (2011) [12]	Case Series 02/2000–06/2009	N = 13 Mean 58.2 y/o Primary unresectable ICC Tumor size 0.9–8 cm	Percutaneous US-guided RFA S = mean 1.3	LTPFS = 32.2 mo OS 1, 3, 5 yr: 85%, 51%, 15% Median OS = 38.5 mo	Major = 6% Minor = 77%
Carrafiello (2010) [24]	Case Series 02/2004–07/2008	N = 6 Mean 69.8 y/o Unresectable ICC Tumor size 1.0–5.8 cm	Percutaneous US-guided RFA S = mean 1.5	Median OS = 20 mo	Major = 0% Minor = 17%
Kamphues (2010) [15]	Retrospective Study 04/2002–05/2008	N = 13 Median 62 y/o Recurrent ICC after SR or RFA Tumor size < 5 cm	IORFA vs. percutaneous US-guided RFA	Mean TTR1: 14 mo Mean TTR2: 14.6 mo OS 1, 3 yr: 92%, 52% Median OS = 51 mo	Major = 7.6%
Chiou (2005) [30]	Case Series 01/2002–10/2004	N = 10 Mean 66.2 y/o Unresectable ICC Tumor size 1.9–6.8 cm	Percutaneous US-guided RFA S = mean 1.2	Total tumor necrosis = 80%	Major = 10% Minor = 0%
** Comparative Cohorts**					
Xiang (2020) [17]	Retrospective Cohort Study 2004–2014	N = 34 RFA N = 150 SR Stage I, Tumor size < 5 cm	RFA vs. SR	OS 1, 3, 5 yr: RFA = 89.9%, 42.4%, 23.9% SR = 87.4%, 73.3%, 61.5% Median OS RFA = 39 mo Median OS SR = 38 mo	
Wu (2019) [31]	Retrospective Cohort Study 2004–2013	N = 86 RFA N = 419 ChR Nonsurgical patients Stage I or II, Tumor size < 5 cm	RFA vs. ChR	5 yr OS Stage I: RFA = 20.1% ChR = 3.7% 3 yr OS Stage II: ND	

Abbreviations: ChR = chemoradiation; CT = computed tomography; DFS = disease-free survival; ICC = intrahepatic cholangiocarcinoma; IORFA = intraoperative radiofrequency ablation; LTPFS = local tumor progression-free survival; mo = month; MWA = microwave ablation; ND = no difference between groups; OS = overall survival; RFA = radiofrequency ablation; RFT = recurrence-free time; S = number of sessions; SFRA = stereotactic radiofrequency ablation; SR = surgical resection; TTR1 = time to 1st recurrence relative to primary surgical resection; TTR2 = time to 2nd recurrence relative to primary surgical resection; US = ultrasound; x = occurrences; yr = years; y/o = years old.

**Table 2 cancers-15-02384-t002:** Studies examining microwave ablation (MWA) for intrahepatic cholangiocarcinoma.

MWA					
Author (Year)	Study Type and Time Period	Patient Population	Technique	Outcomes	Complications
**Single Cohorts**					
Wang (2022) [33]	Retrospective Study 02/2012–12/2020	N = 29 Mean 56.34 y/o Untreated ICC Tumor size 0.5–8.1 cm	Percutaneous US-guided MWA	Median PFS = 18.43 mo Median OS = 18.43 mo	
Kim-Fuchs (2021) [35]	Systematic Review and Retrospective Study 2019–2021	Primary ICC N = 5 Recurrent ICC N = 5 Mean 58.1 y/o Primary and recurrent ICC Tumor size 0.6–3.2 cm	Stereotactic MWA	OS Primary ICC = 6–31.5 mo, all patients still living OS Recurrent ICC = 1–20 mo, 2 patients still living	Dindo IIIa+: 10%
Yang (2021) [36]	Retrospective Study 04/2011–03/2018	N = 55 Mean 59.6 y/o Untreated ICC Tumor size 0.8–5 cm	Percutaneous US-guided MWA	1, 3, 5 yr OS: 87.4%, 51.4%, 35.2% 1, 3, 5 yr RFS: 68.9%, 56.9%, 56.9%	Major = 3.8%
Ni (2019) [39]	Retrospective Study 04/2011–03/2018	N = 78 Mean 59.6 y/o Early-stage, unresectable, and untreated ICC Tumor size < 5 cm	Percutaneous CT-guided MWA	1, 3, 5 yr OS: 89.5%, 52.2%, 35.0% 1, 3, 5 yr RFS: 78.9%, 19.9%, 0%	Major = 3.8% Minor = 29.5%
Zhang (2018) [41]	Retrospective Study 01/2009–02/2016	N = 107 Mean 58 y/o Primary and recurrent ICC Tumor size < 5 cm	Percutaneous US-guided MWA	Median OS = 28.0 mo Median PFS = 8.9 mo	Major = 2.8%
Yang (2015) [42]	Retrospective Study 01/2011–12/2014	N = 26 Mean 57.9 y/o Primary unresectable and recurrent ICC Tumor size 2.5–6.5 cm	Percutaneous US-guided MWA + TACE	Median OS = 19.5 mo Median PFS = 6.2 mo	Major = 0%
Xu (2012) [25]	Retrospective Study 10/1998–08/2010	MWA or RFA N = 18 Mean 60.0 y/o Primary and recurrent ICC after SR Tumor size 0.7–6.9 cm	Percutaneous US-guided MWA or RFA	Median OS = 8.8 mo Median RFS = 4.0 mo	Major = 5.5%
Yu (2011) [44]	Retrospective Study 05/2006–03/2010	N = 15 Mean 57.4 y/o Unresectable ICC Tumor size 1.3–9.9 cm	Percutaneous US-guided MWA	Median OS = 10 mo	Major = 20%
**Comparative Cohorts**					
Ge (2020) [37]	Retrospective Cohort Study 05/2008–12/2015	PMCT N = 92 TACE N = 183 Median 55 y/o Recurrent unresectable ICC Included tumors > 5 cm	US-guided PMCT vs. TACE	OS TACE > PMCT (*p* = 0.041) RFS TACE > PMCT (*p* = 0.047)	
Giorgio (2019) [38]	Retrospective Cohort Study 01/2008–06/2018	MWA N = 35 RFA N = 36 Mean age MWA, RFA = 72, 75 y/o Unresectable ICC Tumor size 2.2–7.2 cm	Percutaneous US-guided MWA vs. RFA	OS MWA > RFA (*p* < 0.005) PFS MWA > RFA (*p* < 0.005)	Major = 0%
Xu (2019) [40]	Retrospective Cohort Study 04/2011–01/2017	MWA N = 56 SR N = 65 Mean age MWA, SR = 54.5, 53.9 y/o Recurrent ICC after initial SR Tumor size 0.8–5 cm	Percutaneous US-guided MWA vs. SR	Median OS MWA = 31.3 mo Median OS SR = 29.4 mo 1, 3, 5 yr OS: MWA = 81.2%, 42.5%, 23.7% SR = 77.4%, 36.4%, 21.8% (*p* = 0.405)	Major MWA = 5.3% Major SR = 13.8% *p* < 0.001
Takahashi (2018) [26]	Retrospective Study 2006–2015	MWA N = 6 RFA N = 44 Mean 62.5 y/o Primary, locally recurrent, and metastatic ICC Mean tumor size = 1.8 cm	Percutaneous US or CT-guided MWA or RFA	Median OS = 23.6 mo Median DFS = 8.2 mo	Major = 0%
Zhang (2013) [43]	Retrospective Cohort Study 01/2007–12/2011	MWA or RFA N = 77 Repeated SR N = 32 Recurrent ICC after SR Tumor size < 5 cm	MWA or RFA vs. repeated SR	Median OS: MWA or RFA = 21.3 mo Repeated SR = 20.3 mo (*p*= 0.996)	Major MWA or RFA = 3.9% Major SR = 46.9% *p* < 0.001
Yan (2022) [34]	Retrospective Cohort Study 01/2010–12/2018	MWA or RFA +ChT N = 55 ChT alone N = 134 Unresectable and untreated ICC Included tumors > 5 cm	RFA or MWA + ChT vs. ChT alone	Median OS: RFA or MWA + ChT = 15.23 mo ChT alone = 7.97 mo *p* = 0.009	

Abbreviations: ChT = chemotherapy; CT = computed tomography; DFS = disease-free survival; ICC = intrahepatic cholangiocarcinoma; mo = month; MWA = microwave ablation; ND = no difference; OS = overall survival; PFS = progression-free survival; PMCT = percutaneous microwave coagulation therapy; RFA = radiofrequency ablation; RFS = recurrence-free survival; SR = surgical resection; TACE = transarterial chemoembolization; US = ultrasound; yr = year; y/o = years old.

**Table 3 cancers-15-02384-t003:** (A) Studies examining transarterial chemoembolization (TACE) and (B) Adjuvant TACE for intrahepatic cholangiocarcinoma.

**(A) TACE**					
**Author (Year)**	**Study Type and Time Period**	**Patient Population**	**Technique**	**Outcomes**	**Complications**
**Single Cohorts**					
Luo (2020) [49]	Prospective Study 11/2015–11/2016	N = 37 Mean 62.9 y/o Primary ICC Tumor size 3–8.3 cm	DEB-TACE	Mean OS = 376 days CR = 8.1% PR = 59.5%	Major = 0%
Zhou (2020) [52]	Retrospective Study 11/2015–05/2018	N = 88 Unresectable ICC Included tumors > 5 cm	DEB-TACE	Median OS = 9.0 mo Median PFS = 3.0 mo	Major = 0%
Schicho (2017) [55]	Prospective Study 01/2010–06/2014	N = 7 Mean = 73.7 y/o Unresectable ICC	DSM-TACE	OR = 12% DC = 44% DP = 4%	Major = 4%
Kim (2008) [56]	Retrospective Study 02/1997–05/2007	N = 49 Mean 62.9 y/o Unresectable ICC Tumor size 1.5–16 cm	TACE and TACI	Median OS = 12 mo 1, 2, 3, 4 yr OS: 46%, 38%, 30%, 15%	Major = 2%
**Comparative Cohorts**					
Sun (2022) [53]	Retrospective Cohort Study 01/2016–06/2020	DEB-TACE N = 40 cTACE N = 49 Mean 59.6 y/o Unresectable ICC Included tumors > 5 cm	DEB-TACE vs. cTACE	DEB-TACE OS = median 10 mo cTACE OS = median 6 mo	DEB-TACE = 12.5% cTACE = 6.1%
Baydoun (2021) [3]	Case Series 01/2013–01/2019	N = 10 Mean 65.3 y/o Primary and recurrent ICC	TACE + RFA vs. TACE vs. RFA	All groups median OS = 29.5 mo All groups median PFS = 15.5 mo	TACE = 0% RFA Major = 0%
Hu (2020) [48]	Retrospective Cohort Study 10/2015–12/2019	N = 35 Mean 57.2 y/o Unresectable stage III and IV ICC Included tumors > 5 cm	Ap + DEB-TACE vs. Ap + cTACE vs. Ap	Ap + DEB-TACE median OS = 19.3 mo Ap + cTACE median OS = 14.0 mo Ap median OS = 6.5 mo	Ap + DEB-TACE ≈ Ap + cTACE > Ap
Park (2011) [6]	Retrospective Cohort Study 01/1996–04/2009	TACE N = 72 Supportive care N = 83 Mean 64.6 y/o Unresectable ICC Included tumors > 5 cm	TACE vs. supportive care	Median OS: TACE = 12.2 mo Supportive care = 3.3 mo	Grade 3 heme AE = 13% Grade 3 non-heme AE = 24%
Poggi (2009) [57]	Retrospective Cohort Study 12/2005–05/2008	ChT + OEM-TACE N = 9 ChT N = 11 Unresectable ICC	ChT + OEM-TACE vs. ChT	Median OS: ChT + OEM-TACE = 30 mo ChT = 12.7 mo	Grade 4 complications = 0%
**(B) Adjuvant TACE**					
**Author (Year)**	**Study Type and Time Period**	**Patient Population**	**Technique**	**Outcomes**	**Complications**
Cheng (2021) [58]	Retrospective Cohort Study 12/2002–11/2015	SR + TACE N = 68 SR alone N = 155 Mean 51.8 y/o Included tumors > 5 cm	SR + adjuvant TACE vs. SR alone	SR + TACE vs. SR alone: 1, 3, 5 yr OS = ND 1, 3, 5 yr RR = ND	
Li (2015) [59]	Retrospective Cohort Study 01/2008–02/2011	SR + TACE N = 122 SR alone N = 431 Mean 54 y/o Included tumors > 5 cm	SR + adjuvant TACE vs. SR alone	SR + TACE vs. SR alone: 1, 3, 5 yr OS = ND 1, 3, 5 yr RR = ND	
Wu (2012) [31]	Retrospective Cohort Study 01/2005–12/2006	SR + TACE N = 57 SR alone N = 57 Median 56 y/o Included tumors > 5 cm	SR + adjuvant TACE vs. SR alone	Poor prognostic factors: 1, 3, 5 yr OS = SR + TACE > SR (*p* <0.001) Without poor prognostic factors: 1, 3, 5 yr OS = ND	
Shen (2011) [60]	Retrospective Cohort Study 07/2002–12/2003	SR + TACE N = 53 SR alone N = 72 Included tumors > 5 cm	SR + adjuvant TACE vs. SR alone	1, 3, 5 yr OS = SR + TACE > SR (*p* <0.045) Median OS in patients with early recurrence: SR + TACE = 12 mo SR alone = 5 mo	Major = 0%

Abbreviations: AE = adverse effect; Ap = apatinib; ChT = chemotherapy; CR = complete response; cTACE = conventional TACE; DEB-TACE = drug-eluting bead TACE; DC = disease control; DP = disease progression; DSM-TACE = degradable starch microsphere TACE; heme = hematologic; ICC = intrahepatic cholangiocarcinoma; mo = months; ND = no difference; OEM-TACE = oxaliplatin-eluting microsphere TACE; OR = objective response; OS = overall survival; PFS = progression-free survival; PR = partial response; RFA = radiofrequency ablation; RR = recurrence rate; SR = surgical resection; TACE = transarterial chemoembolization; TACI = transarterial chemoinfusion; TARE = transarterial radioembolization; yr = year; y/o = years old.

**Table 4 cancers-15-02384-t004:** Studies examining transarterial radioembolization (TARE) for intrahepatic cholangiocarcinoma.

TARE					
Author (Year)	Study Type and Time Period	Patient Population	Technique	Outcomes	Complications
**Single Cohorts**					
Paprottka (2021) [62]	Retrospective Study	N = 73 Mean 64.5 y/o Unresectable ICC Included tumor burden >50%	TARE Y90 resin microspheres	Mean OS = 18.9 mo Mean PFS = 10.1 mo	Grade 3: 12.3% Grade 4–5: 0%
Paz-Fumagalli (2021) [63]	Retrospective Study	N = 28Mean 64.2 y/o Unresectable ICC Tumor size 2–14 cm	TARE Y90 glass microspheres	1, 3 yr OS: 78%, 59% Median PFS = 8.8 mo	Grade 3+: 7.1%
Edeline (2020) [66]	Phase II Clinical Trial 11/2013–06/2016	N = 41 Mean 64 y/o Unresectable and recurrent ICC Included tumors > 2 cm	SIRT + ChT Y90 glass microspheres	Median OS = 22 mo Median PFS = 14 mo	Grade 3–4: 71%
Filippi (2019) [67]	Case Series	N = 9 Mean 65.4 y/o Recurrent ICC after 1st TARE	Repeat TARE Y90 resin microspheres	Median OS = 16.5 mo after 1st TARE	Grade 3+: 0%
Kohler (2019) [68]	Retrospective Study	N = 46 Median 62.5 y/o Advanced and recurrent ICC Included tumor burden > 50%	TARE Y90 resin microspheres	Median OS = 9.5 mo	
Levillain (2019) [69]	Retrospective Study 01/2004–9/2018	N = 58 Median 66 y/o Unresectable, ChT refractory ICC Included tumors > 2 cm	TARE Y90 resin microspheres	Median OS = 10.3 mo 1, 2 yr OS: 40%, 22%	
White (2019) [70]	Prospective Study 12/2013–02/2017	N = 61 Median 64 y/o ChT refractory ICC	TARE Y90 resin or glass microspheres	Median OS = 8.7 mo Median PFS = 2.8 mo	Grade 3+: 8%
Gangi (2018) [71]	Retrospective Study 05/2009–05/2016	N = 85 Mean 73.4 y/o Unresectable ICC	TARE Y90 glass microspheres	Median OS = 12 mo	Grade 3+: 8.2%
Shaker (2018) [72]	Retrospective Study 2006–2016	N = 17 Median 69.3 y/o Unresectable and metastatic ICC, Stage I-IV Mean tumor size = 7.4 cm	TARE Y90 resin or glass microspheres	Median OS = 33.6 mo Median PFS = 4 mo	Technical complications: 9%
Jia (2017) [74]	Retrospective Study 02/2006–09/2015	N = 24 Mean 61.8 y/o Unresectable and ChT refractory ICC	TARE Y90 resin microspheres	Median OS from Dx, ChT, and TARE: 24 mo, 16 mo, 9 mo	Grade 3+: 20.8%
Mosconi (2016) [75]	Retrospective Study 07/2010–09/2015	N = 23 Mean 65 y/o Unresectable and recurrent ICC	TARE Y90 resin microspheres	Median OS = 17.9 mo 1, 2 yr OS: 67.9%, 20.6%	Grade 3: 8.7%
Mouli (2013) [76]	Retrospective Study 07/2003–05/2011	N = 46 Median 68 y/o Unresectable ICC Included tumor burden > 50%	TARE	Median OS solitary tumor = 14.6 mo Median OS multifocal tumors = 5.7 mo	Grade 3+: 17%
Hoffmann (2012) [77]	Retrospective Study 04/2007–01/2010	N = 33 Mean 65.2 y/o Unresectable ICC Included tumor burden < 50%	TARE Y90 resin microspheres	Median OS = 22 mo Median PFS = 9.8 mo	Major = 0%
Saxena (2010) [78]	Retrospective Study 01/2004–05/2009	N = 25 Mean 57 y/o Unresectable ICC Included tumor burden <50%	TAREY90 resin microspheres	Median OS = 9.3 mo 6 mo, 1 yr, 2 yr, 3 yr OS: 56%, 40%, 27%, 13%	Grade 3+: 24%
**Comparative Cohorts**					
Bargellini (2020) [64]	Retrospective Cohort Study 07/2008–10/2017	N = 81 Mean 62.4 y/o Unresectable ICC Mean tumor size = 5.98 cm	A = ChT naïve + TARE B = ChT + adjuvant TARE C = progression after ChT + TARE	Median OS = 14.5 mo (ND between groups)	Major = 0%
Buettner (2020) [65]	Retrospective Cohort Study 06/2006–02/2017	Resin N = 92 Glass N = 22 Unresectable ICC Tumor size 5.4–10 cm	TARE Resin vs. glass microspheres	Median OS = 29 mo (ND between groups) Median PFS = 5 mo (ND between groups)	Grade 3: 4%
Akinwande (2017) [73]	Retrospective Cohort Study 08/2001–07/2016	TARE N = 25 TACE N = 15 Median age TARE = 64 y/o Median age TACE = 60 y/o Unresectable ICC Included tumor burden >50%	TARE vs. TACE		Grade 3+ TARE: 10% Grade 3+ TACE: 9% (ND)

Abbreviations: ChT = chemotherapy; Dx = diagnosis; ICC = intrahepatic cholangiocarcinoma; mo = month; ND = no difference; OS = overall survival; PFS = progression-free survival; SIRT = selective internal radiotherapy; TACE = transarterial chemoembolization; TARE = transarterial radioembolization; yr = year; y/o = years old; Y90 = yttrium-90.

**Table 5 cancers-15-02384-t005:** Studies examining external body radiotherapy (EBRT) for intrahepatic cholangiocarcinoma.

EBRT					
Author (Year)	Study Type and Time Period	Patient Population	Technique	Outcomes	Complications
**Single Cohorts**					
Smart (2020) [81]	Retrospective Study 2008–2018	N = 66 Median 76 y/o Unresectable or locally recurrent ICC Tumor size 2.5–16 cm	Hypofractionated proton or photon EBRT Median RT dose = 58.05 Gy	2 yr LC = 84% 2 yr OS = 58% Median OS = 25 mo	Grade 3+: 11%
Tao (2016) [84]	Retrospective Study 2002–2014	EBRT + ChT N = 70 EBRT alone N = 9 Unresectable ICC, stage I-IV Tumor size 2.2–17 cm	EBRT Median RT dose = 58.05 Gy Median BED = 80.5 Gy	Median OS = 30 mo 3 yr OS: BED > 80.5 Gy = 73% BED < 80.5 Gy = 38%	6.3% hospitalized within 90 days of EBRT
**Comparative Cohorts**					
Kolarich (2018) [85]	Retrospective Study NCDB 2004–2015	N = 2222 EBRT, RFA, RI, and no local therapy Nonsurgical patients, stage I-IV ICC	EBRT, RFA, or RI vs. no local therapy	Stage I median OS: RFA = 2.1 yr, EBRT = 1.7 yr, No local therapy = 0.7 yr Stage II median OS: ND Stage III median OS: EBRT = 0.9 yr, RI = 1.2 yr, No local therapy = 0.6 yr Stage IV median OS: RI = 0.9 yr, No local therapy = 0.3 yr	
Shao (2018) [80]	Retrospective Cohort Study SEER 1973–2013	Palliative EBRT N = 847 No EBRT = 3180 Median 64 y/o Palliative EBRT for unresectable ICC Included tumors > 5 cm	Palliative EBRT vs. no EBRT	OS: EBRT > none (HR = 0.844, *p* = 0.00228) CSS: EBRT > none (HR = 0.8563, *p* = 0.0037)	
Hammad (2016) [86]	Retrospective Cohort Study NCDB 1998–2013	Total N = 2897 EBRT N = 525 Median 65 y/o Adjuvant EBRT Included tumors > 5 cm	SR + EBRT vs. SR alone	Median OS R1/R2 LN (-) patients: SR + EBRT = 39.5 mo SR alone = 21.1 mo *p* = 0.052	
Jackson (2016) [87]	Retrospective Study NCDB 2001–2011	Total N = 1636 Median 63 y/o Unresectable, localized ICC	EBRT + ChT vs. ChT alone	2 yr OS: EBRT + ChT = 25.8% ChT alone = 20% *p* = 0.001	
Chen (2010) [82]	Retrospective Cohort Study 12/1998–12/2008	Palliative EBRT N = 35 No EBRT N = 49 Palliative EBRT for unresectable ICC, stage I-IV Included tumors > 10 cm	EBRT vs. no EBRT Median RT dose = 50 Gy	Median OS: EBRT = 9.5 mo No EBRT = 5.1 mo 1, 2 yr OS: EBRT = 38.5%, 9.6% No EBRT = 16.4%, 4.9%	Grade 3: 11.4% 1 RILD resulting in mortality
Jiang (2010) [88]	Retrospective Cohort Study 01/1999–12/2008	EBRT N = 24 No EBRT N = 66 Resected ICC with LN metastasis Included tumors > 10 cm	EBRT vs. no EBRT Median RT dose = 50 Gy	Median OS: EBRT = 19.1 mo No EBRT = 9.5 mo	Grade 3: 12.5%
Shinohara (2008) [89]	Retrospective Cohort Study SEER 1988–2003	Total N = 3839 Median 73 y/o Adjuvant and definitive EBRT	SR + EBRT or BI vs. SR alone vs. EBRT or BI alone vs. no treatment	Median OS: SR + EBRT/BI = 11 mo SR alone = 6 mo EBRT/BI alone = 7 mo No treatment = 3 mo	
Zeng (2006) [83]	Retrospective Cohort Study 01/1998–12/2004	EBRT N = 38 No EBRT N = 37 Unresectable ICC and post-op EBRT for LN (+) metastasis Included tumors > 10 cm	EBRT vs. no EBRT Median RT dose = 50 Gy	1, 2 yr OS: EBRT = 50.1%, 11.8% No EBRT = 24.8%, 5.5% *p* = 0.005	Grade 3: 7.9%

Abbreviations: BED = biologic equivalent dose; BI = brachytherapy implants; ChT = chemotherapy; CSS = cancer-specific survival; EBRT = external body radiotherapy; ICC = intrahepatic cholangiocarcinoma; LC = local control; LN = lymph node; mo = month; NCDB = National Cancer Database; ND = no difference; OS = overall survival; RFA = radiofrequency ablation; RI = radioactive implant; RILD = radiation-induced liver disease; RT = radiation therapy; R1/R2 = positive margins after surgical resection; SEER = The Surveillance, Epidemiology, and End Results database; SR = surgical resection; s/p = status-post; yr = year; y/o = years old.

**Table 6 cancers-15-02384-t006:** Studies examining stereotactic body radiotherapy (SBRT) for intrahepatic cholangiocarcinoma.

SBRT					
Author (Year)	Study Type and Time Period	Patient Population	Technique	Outcomes	Complications
**Single Cohorts**					
Kozak (2020) [97]	Retrospective Study 2003–2017	ICC N = 25 PHCC N = 15 Median 71 y/o Adjuvant and definitive SBRT Tumor size 1–12.5 cm	SBRT Median RT dose = 40 Gy	Median OS ICC = 23 mo Median OS PHCC = 10 mo *p* = 0.018 1 + 2 yr cumulative RF incidence: ICC = 8% PHCC = 24%	Acute Grade 3–4: 42.5% Acute Grade 5: 2.5% Late Grade 3–4: 45% Late Grade 5: 0%
Brunner (2019) [98]	Retrospective Study 07/1999–09/2016	Total N = 64 Median 64 y/o Locally advanced ICC and ECC Tumor size 1–18 cm	SBRT Median BED = 62.7 Gy	Median OS = 15 mo 1, 2 yr OS: 61%, 34%	Grade 3 GI bleed: 4.7% Grade 4–5: 0%
Gkika (2017) [99]	Retrospective Study 2007–2016	ICC tumor N = 17 ECC tumor N = 26 Unresectable, positive margins after SR and recurrent disease Tumor size 2–18 cm	SBRT Median RT dose = 45 Gy	Median OS = 14 mo Median PFS = 9 mo	Grade 3 bleeding: 9%
Shen (2017) [100]	Retrospective Study	N = 28 Unresectable ICC Included tumors > 10 cm	SBRT Median RT dose = 45 Gy	Median OS = 15 mo Median PFS = 11 mo	Grade 3: 53.6% Grade 4–5: 0%
Sandler (2016) [95]	Retrospective Study 10/2008–6/2015	ICC N = 6 ECC N = 25 Median 63 y/o Locally advanced disease Tumor size 1–7.3 cm	SBRT Median RT dose = 40 Gy	Median OS = 15.7 mo Median PFS = 16.8 mo	Grade 1–2: 77% Grade 3+: 16%
Weiner (2016) [94]	Phase I/II Prospective Study 02/2012–05/2014	HCC N = 12 ICC N = 12 Biphenotypic tumor N = 2 Median 72 y/o Unresectable disease Tumor size 1.6–12.3 cm	SBRT Median RT dose = 55 Gy	HCC median OS = 9.8 mo ICC/biphenotypic tumor median OS = 13.2 mo HCC median PFS = 5.3 mo ICC/biphenotypic tumor median PFS = 24.7 mo	Grade 4–5: 11%
Mahadevan (2015) [91]	Retrospective Study 02/2006–02/2014	ICC N = 31 Hilar CC = 11 Median 72 y/o Unresectable disease or positive margins after SR	SBRT Median RT dose = 30 Gy	Median OS = 17 mo Median PFS = 10 mo	Grade 3: 12% Grade 4–5: 0%
Barney (2012) [92]	Retrospective Study 03/2009–07/2011	ICC N = 6 ECC N = 4 Median 61.6 y/o Primary or recurrent disease	SBRT Median RT dose = 55 Gy	OS 6 mo, 1 yr: 83%, 73% Median PFS = 6.1 mo	Grade 3: 10% Grade 4–5: 10%
Tse (2008) [96]	Phase I Clinical Trial 08/2003–03/2006	ICC N = 10 HCC N = 31 Median 62 y/o Unresectable disease Tumor size 9–1913 mL	SBRT Median RT dose = 36 Gy	ICC median OS = 15 mo HCC median OS = 11.7 mo	Grade 3: 43.9% Grade 45: 0%
**Comparative Cohorts**					
Sebastian (2019) [101]	Retrospective Cohort Study NCDB 2004–2014	SBRT N = 27 ChR N = 54 TARE N = 60 Unresectable ICC Tumor size 2.9–9.2 cm	SBRT vs. ChR vs. TARE Median RT dose SBRT = 45 Gy	OS with propensity weighting: SBRT > ChR (*p* < 0.0001) SBRT vs. TARE = NS (*p* = 0.019)	

Abbreviations: BED = biological effective dose; ChR = chemoradiation; ECC = extrahepatic cholangiocarcinoma; GI = gastrointestinal; HCC = hepatocellular carcinoma; ICC = intrahepatic cholangiocarcinoma; mo = months; NCDB = National Cancer Database; NS = not significant; OS = overall survival; PFS = progression-free survival; PHCC = perihilar cholangiocarcinoma; RF = regional failure; RT = radiation therapy; SBRT = stereotactic body radiotherapy; SR = surgical resection; TARE = transarterial radioembolization; yr = year; y/o = years old.

**Table 7 cancers-15-02384-t007:** Studies examining hepatic arterial infusion (HAI) for intrahepatic cholangiocarcinoma.

HAI					
Author (Year)	Study Type and Time Period	Patient Population	Technique	Outcomes	Complications
**Single Cohorts**					
Huang (2022) [107]	Retrospective Study 12/2020–05/2021	N = 9 Mean age 55.3 y/o ChT and immunotherapy refractory, unresectable ICC Tumor size 3.3–13 cm	HAI = FOLFIRI Mean # HAI cycles = 2.9	6 mo OS: 22.2% Median PFS = 5 mo	Grade 3–4: 22%
Pietge (2021) [112]	Prospective Study 06/2012–01/2016	N = 12 ICC, Hilar CC and gallbladder cancer Median age 63.5 y/o	HAI = FUDR + Systemic ChT = GC	Median OS = 23.9 mo Median PFS = 10.1 mo	Serious adverse event N = 16
Cercek (2020) [113]	Phase II Clinical Trial 05/2013–05/2019	N = 38 Median age 64 y/o Unresectable ICCTumor size 1.7–24.8 cm	HAI = FUDR + Systemic ChT = GEMOX	Median OS = 25.0 mo Median PFS = 11.8 mo	Grade 4 requiring removal from study: 11%
Kasai (2014) [118]	Retrospective Study 10/2008–07/2013	N = 20 Mean age 62.45 y/o Advanced ICC Tumor size 5.8–19 cm	HAI = 5-fluorouracil + SubQ PEG-IFNα-2b	Median OS = 14.6 mo Median PFS = 8.0 mo	Grade 4: 0%
Inaba (2011) [119]	Phase I/II Clinical Trial 05/2004–11/2006	N = 25 Median 58 y/o Unresectable ICC	HAI = gemcitabine	Median OS = 340 days	Grade 4: 4%
Kemeny (2011) [120]	Retrospective Study	ICC N = 18 HCC N = 4 Unresectable ICC or HCC Tumor size 1.1–16.4 cm	HAI = FUDR + dexamethasone + Systemic Bev	Median OS = 31.1 mo Median PFS = 8.45 mo	Grade 3–4 events N = 32
Jarnagin (2009) [121]	Phase II Clinical Trial 08/2003–03/2007	ICC N = 26 HCC N = 8 Mean age 56.5 y/o Unresectable ICC or HCC Tumor size 2.7–18.1 cm	HAI = FUDR + dexamethasone	Median OS = 29.5 mo Median PFS = 7.4 mo	Grade 3–4: 14.7%
**Comparative Cohorts**					
Franssen (2022) [106]	Retrospective Cohort Study 01/2001–12/2018	HAI N = 141 SR N = 178 Median age HAI 62 y/o Median age SR 60 y/o Multifocal ICC Median tumor size HAI, SR = 8.4 cm, 7.0 cm	HAI = FUDR vs. SR Median # HAI cycles = 8	Median OS HAI = 20.3 mo Median OS SR = 18.9 mo (*p* = 0.32)	HAI Grade 3A+: 6.4% SR Grade 3A+: 25.3% (*p* = 0.04)
Ishii (2022) [108]	Retrospective Cohort Study 04/2014–12/2020	HAI N = 18 ChT N = 24 Mean age 64 y/o Advanced ICC Tumor size 1–12.1 cm	HAI = GEM-FP vs. Systemic ChT = GC	Median OS HAI = 19.7 mo Median OS ChT = 10.8 mo (*p* = 0.006)	HAI vs. ChT = ND except leukopenia > in ChT group
Zhang (2022) [109]	Retrospective Cohort Study 01/2021–03/2022	HAI N = 39 TACE N = 19 Unresectable ICC Included tumors >7 cm	HAI = GEMOX vs. TACE	Median PFS HAI = not reached by end of study Median PFS TACE = 11 mo	HAI vs. TACE = ND
Cai (2021) [110]	Retrospective Cohort Study 03/2011–10/2019	HAI N = 57 TACE N = 69 Unresectable ICC Included tumors >5 cm	HAI = mFOLFOX vs. TACE	Median OS HAI = 19.6 mo Median OS TACE = 10.8 mo Median PFS HAI vs. TACE = ND	Total HAI N = 26 Total TACE N = 9
Jolissaint (2021) [111]	Retrospective Cohort Study 2008–2018	HAI N = 196 SR N = 237 ChT N = 140 ICC with LN metastasis Included tumors >10 cm	HAI = FUDR vs. SR vs. Systemic ChT	LN (-) median OS: SR = 59.9 mo, HAI = 24.9 mo, ChT = 13.7 mo (*p* < 0.001) LN (+) median OS: SR vs. HAI = ND	
Higaki (2018) [114]	Retrospective Cohort Study 2007–2011	HAI + ChT N = 12 Other N = 16 Unresectable ICC Median age HAI + ChT = 76 y/o Median age Other = 67 y/o Included tumors > 5 cm	HAI + Systemic ChT vs Other treatment (radiation, TACE or systemic ChT alone)	Median OS HAI + ChT = 10.1 mo Median OS Other = 4.0 mo	Grade 3–4: 4.54%
Wright (2018) [115]	Retrospective Cohort Study 01/2004–06/2016	IAT N = 59 SR N = 57 Mean age IAT 61.9 y/o Mean age SR 64.9 y/o Multifocal ICC Included tumors > 10 cm	IAT = HAI, TACE or TARE vs. SR	Median OS IAT = 16 mo (HA = 39 mo, TACE = 15 mo) Median OS SR = 20 mo	
Konstantinidis (2016) [116]	Retrospective Cohort Study 01/2000–08/2012	HAI + ChT N = 78 ChT alone N = 26 Median age 62 y/o Unresectable ICC Tumor size 1.5–16.4 cm	HAI = FUDR + Systemic ChT vs. Systemic ChT alone	Median OS HAI + ChT = 30.8 mo Median OS ChT alone = 18.4 mo	
Konstantinidis (2014) [117]	Retrospective Cohort Study 08/2003–09/2009	N = 44 Mean 59 y/o Unresectable ICC Mean tumor size = 9.3 cm	HAI = FUDR vs. FUDR + Bev	Median OS FUDR = 29.3 mo Median OS FUDR + Bev = 28.5 mo	Grade 3–4: 22.7%

Abbreviations: Bev = bevacizumab; CC = cholangiocarcinoma; ChT = chemotherapy; FOLFIRI = 5-fluorouracil + irinotecan; FUDR = floxuridine; GC = gemcitabine + cisplatin; GEM-FP = gemcitabine + cisplatin + 5-fluorouracil; GEMOX = gemcitabine + oxaliplatin; HAI = hepatic arterial infusion; HCC = hepatocellular carcinoma; IAT = intra-arterial therapy; ICC = intrahepatic cholangiocarcinoma; LN = lymph node; mFOLFOX = leucovorin + fluorouracil + oxaliplatin; mo = month; ND = no difference; OS = overall survival; PFS = progression-free survival; SR = surgical resection; SubQ = subcutaneous; TACE = transarterial chemoembolization; TARE = transarterial radioembolization; yr = year; y/o = years old.

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
