# Peer review of "Locoregional Therapy for Intrahepatic Cholangiocarcinoma"

_cancers, 2023, doi:10.3390/cancers15082384_

Round 1

Reviewer 1 Report

excellent literature review on a very controversial topic. very well written article

Author Response

Thank you for your kind comment.

Reviewer 2 Report

This is a well-written review paper with promising guidance on the clinical management of Intrahepatic Cholangiocarcinoma.

Author Response

Thank you for your kind comment.

Reviewer 3 Report

The review focuses on locoregional treatment methods for intrahepatic cholangiocarcinoma with the description of each method and overviews in seperate tables.

As there are many studies with only small patient groups, it is difficult to compare them or draw conclusions. Therefore, a review cannot give substantial new information.

I would recommend to add brachytherapy as a local treatment.

In a methods section, methods for this review need to be explained. 

For the tables, a better separation of single cohort or comparative cohorts is advisable.

Important information for the tables need to be: 1. local control after 1y, 2y,...2. if available, also distant control or generally PFS; 3. toxicity uniformly always reported in grades (not major or minor); 4. oucome date specified (not just "OS", but specifically e.g. "median OS"

Considering radiotherapy, the concept of a "biologically equivalent dose" should be mentioned to explain different dose levels. To compare dose concepts, not only total doses, but dose per fraction is relevant. The differentiation between EBRT and SBRT cmainly depends on dose per fraction. Please, comment.

In the discussion or conclusions, an improved comparison of pros and cons for the decesion of a specific method would be helpful.

Author Response

The review focuses on locoregional treatment methods for intrahepatic cholangiocarcinoma with the description of each method and overviews in seperate tables. As there are many studies with only small patient groups, it is difficult to compare them or draw conclusions. Therefore, a review cannot give substantial new information.

We agree that there are many studies with small patient groups and that it is therefore difficult to compare them or draw conclusions. Unfortunately, intrahepatic cholangiocarcinoma is rare and patients often present with disease that is metastatic and not amenable to local or locoregional therapy. Our goal with this narrative review was to summarize and condense the available literature to help guide providers. We have added a statement about the narrative nature of this review.

I would recommend to add brachytherapy as a local treatment.

We have added a short section on brachytherapy as a local treatment.

In a methods section, methods for this review need to be explained. 

This is a narrative review and narrative reviews do not typically include methods sections. We therefore did not include one. However, in response to the reviewer’s comments we have added this short section following the introduction:

In this narrative review, we summarized articles related to locoregional therapies for intrahepatic cholangiocarcinoma. We included articles published between 2005 and 2022. We included articles with data on outcomes from the following locoregional therapies: RFA, MWA, TACE, TARE, EBRT, SBRT, HAI, IE and brachytherapy. Articles were excluded if they focused on extrahepatic cholangiocarcinoma including distal or hilar or cholangiocarcinoma, or gallbladder cancer, or if they included patients with all biliary tract cancers and the data for patients with intrahepatic cholangiocarcinoma could not be separated.

For the tables, a better separation of single cohort or comparative cohorts is advisable.

Important information for the tables need to be:

  1. local control after 1y, 2y,...2. if available, also distant control or generally PFS;

Information about local control, OS, PFS was reported as provided in each individual paper. Because studies were small and not uniform, different outcome measures were reported for each study and these have been included.

  1. toxicity uniformly always reported in grades (not major or minor);

Similarly, toxicity was reported as described in each individual papers. Some papers reported toxicity in terms of self-defined “major vs minor” complications while others reported toxicity in terms of grades. Again, this was reported in the table as it was in the original paper and varied between papers.

  1. outcome date specified (not just "OS", but specifically e.g. "median OS"

Outcome data was reported as it was originally in the cited sources. OS was specified as mean or median OS when available.

Considering radiotherapy, the concept of a "biologically equivalent dose" should be mentioned to explain different dose levels.

To compare dose concepts, not only total doses, but dose per fraction is relevant.

The differentiation between EBRT and SBRT cmainly depends on dose per fraction.

We included data as it was provided in the originally cited articles and therefore included data as it was provided in the original articles cited.

Please, comment.In the discussion or conclusions, an improved comparison of pros and cons for the decesion of a specific method would be helpful.

As you mentioned in your first comment: “there are many studies with only small patient groups, it is difficult to compare them or draw conclusions”. Therefore, the pros and cons of each treatment type are discussed in general terms in the “technique” subsection for each treatment and outcomes/limitations of notable studies are highlighted in the “outcomes” subsection. Again, like you mentioned, with the small sample size of studies and variable study design, it is not feasible to make a recommendation about the specific decision for therapy, especially in comparison to each other. This review is intended to summarize and condense available literature, which is sparse.

Round 2

Reviewer 3 Report

The manuscript has been well improved after revision.

The formatting of the tables needs to be corrected before publication to provide a good overview (column sizes etc.).